# Heightened Epstein-Barr virus immunity and potential cross-reactivities in multiple sclerosis

Olivia G. Thomas[1]*, Tracey A. Haigh[2], Deborah Croom-Carter[1], Alison Leese[3], Yolanda Van Wijck[1], Michael R. Douglas[4,5], Alan Rickinson[1], Jill M. Brooks[1‡]*, Graham S. Taylor[2‡]*

1 Institute of Cancer and Genomic Sciences, College of Medical and Dental Sciences, University of Birmingham, Edgbaston, United Kingdom, 2 Institute of Immunology and Immunotherapy, College of Medical and Dental Sciences, University of Birmingham, Edgbaston, United Kingdom, 3 School of Biological Sciences, University of Birmingham, Edgbaston, United Kingdom, 4 Dudley Group of Hospitals NHS Foundation Trust, Dudley, United Kingdom, 5 Institute of Inflammation and Ageing, College of Medical and Dental Sciences, University of Birmingham, Birmingham, United Kingdom

‡ These authors share last authorship on this work.
* olivia.thomas@ki.se (OGT); j.m.brooks@bham.ac.uk (JMB); g.s.taylor@bham.ac.uk (GST)

**Data Availability Statement:** All data are in the manuscript and supporting information files.

**Funding:** This work was funded by the Wellcome Trust (grant ref: WT093580MA to ABR). OGT was

## Abstract

### Background

Epstein-Barr virus (EBV) is a likely prerequisite for multiple sclerosis (MS) but the underlying mechanisms are unknown. We investigated antibody and T cell responses to EBV in persons with MS (pwMS), healthy EBV-seropositive controls (HC) and post-infectious mononucleosis (POST-IM) individuals up to 6 months after disease resolution. The ability of EBV-specific T cell responses to target antigens from the central nervous system (CNS) was also investigated.

### Methods

Untreated persons with relapsing-remitting MS, POST-IM individuals and HC were, as far as possible, matched for gender, age and *HLA-DRB1*15:01*. EBV load was determined by qPCR, and IgG responses to key EBV antigens were determined by ELISA, immunofluorescence and Western blot, and tetanus toxoid antibody responses by multiplex bead array. EBV-specific T cell responses were determined *ex vivo* by intracellular cytokine staining (ICS) and cross-reactivity of *in vitro*-expanded responses probed against 9 novel Modified Vaccinia Ankara (MVA) viruses expressing candidate CNS autoantigens.

### Results

EBV load in peripheral blood mononuclear cells (PBMC) was unchanged in pwMS compared to HC. Serologically, while tetanus toxoid responses were unchanged between groups, IgG responses to EBNA1 and virus capsid antigen (VCA) were significantly elevated (EBNA1 *p = 0.0079*, VCA *p = 0.0298*) but, importantly, IgG responses to EBNA2 and the EBNA3 family antigens were also more frequently detected in pwMS (EBNA2 *p = 0.042* and

supported by a studentship from the Multiple Sclerosis Society (grant ref: 981/12 to JMB). The funders played no role in the study design, data collection, analysis, decision to publish or preparation of the manuscript.

**Competing interests:** The authors have declared that no competing interests exist.

EBNA3 $p = 0.005$). In *ex vivo* assays, T cell responses to autologous EBV-transformed B cells and to EBNA1 were largely unchanged numerically, but significantly increased IL-2 production was observed in response to certain stimuli in pwMS. EBV-specific polyclonal T cell lines from both MS and HC showed high levels of autoantigen recognition by ICS, and several neuronal proteins emerged as common targets including MOG, MBP, PLP and MOBP.

## Discussion

Elevated serum EBV-specific antibody responses in the MS group were found to extend beyond EBNA1, suggesting a larger dysregulation of EBV-specific antibody responses than previously recognised. Differences in T cell responses to EBV were more difficult to discern, however stimulating EBV-expanded polyclonal T cell lines with 9 candidate CNS autoantigens revealed a high level of autoreactivity and indicate a far-reaching ability of the virus-induced T cell compartment to damage the CNS.

### Author summary

Previous infection with Epstein-Barr virus (EBV) is likely required for multiple sclerosis (MS) which occurs when the immune system damages the brain. How EBV contributes to this process is unknown, but some research suggests that immune responses to EBV may react to proteins in the brain that mimic virus proteins. We investigated T-cell and antibody responses to EBV in people with MS (pwMS) and, as controls, people persistently infected with EBV who acquired the virus either silently or symptomatically. pwMS had more frequent antibody responses to several viral proteins, including EBNA1, EBNA2, EBNA3A, EBNA3B and VCA, but their EBV viral load was similar. T cell responses to EBV-transformed B-cells and EBNA1 were only slightly different in pwMS compared to controls. However, T-cell lines from pwMS and healthy individuals, established using each person's own EBV-infected B-cell line, could also respond to multiple brain proteins. T-cell responses to myelin oligodendrocyte glycoprotein (MOG) were particularly common and we isolated T-cells that cross-recognised MOG and EBNA1. Our findings suggest that adaptive immune responses to EBV–which would normally fight EBV infection–are dysregulated in MS and can also target multiple proteins in the brain, indicating their likely involvement in disease.

## Introduction

Multiple sclerosis (MS) is a chronic inflammatory disease of the central nervous system (CNS). MS incidence has been steadily rising in recent decades with current estimates indicating that there are 2.8 million people living with the disease worldwide [1]. MS is thought to be caused by aberrant adaptive immune responses which cross the blood-brain barrier and target myelin sheaths surrounding neurons [2]. Causes of disease have yet to be fully elucidated but are thought to include genetic, environmental and lifestyle factors. The strongest genetic link is with *HLA-DR*15:01*, suggesting that disease pathology involves presentation of antigens via human leukocyte antigen (HLA) class II and CD4+ T cell responses [3]. However, the apparent lack of shared autoantigenic targets between different MS patients has led to much debate as to

the identity of pathogenic responses and, whilst detectable myelin-specific T cells are not restricted to persons with MS, differences in their frequency and avidity have been reported in some studies [4].

EBV, a γ-Herpesvirus, is best known for its association with a range of human tumours, many of these being of B cell origin, and for its ability to immortalise human B cells *in vitro* into permanent lymphoblastoid cells lines (LCLs) constitutively expressing a small set of EBV latent cycle antigens. Yet this potentially pathogenic agent is widespread in all human populations, is commonly acquired in childhood as a silent infection and is carried by most individuals asymptomatically for life. If primary infection is delayed until adolescence or later it may cause infectious mononucleosis (IM), a self-limiting lymphoproliferative disease characterised by an exuberant immune response with large expansions of virus-specific CD8+ T cells alongside smaller responses in the CD4+ T cell and B cell compartments [5]. Remarkably, epidemiological evidence now strongly suggests a role for the virus in MS pathogenesis [6,7]. Firstly, a history of IM is a known risk factor for MS; secondly longitudinal studies have shown that, in the general population, EBV seroconversion almost always precedes disease development [7,8]; thirdly, neurofilament light chain–a known marker of neuroaxonal degeneration–is only elevated following EBV acquisition [6]. Whilst other genetic and environmental factors are also involved, the reported correlation of disease risk with IgG responses to specific regions of the EBV-coded latent cycle nuclear antigen EBNA1 support the theory that it is the adaptive immune response to EBV–rather than infection itself–which is responsible for driving disease [3,9]. How EBV-induced immune responses might contribute to the destruction of CNS tissue observed in MS remains to be determined.

Most work on EBV-specific antibody responses has focused on elevated EBNA1 IgG and studies have reported that these are linked to an 8-fold increase in MS risk [10,11]. Furthermore, EBNA1 IgG antibodies have been detected in oligoclonal bands of MS patients [12] and were recently shown to cross-react with the human proteins anoctamin-2 (ANO2), glial cell adhesion molecule (GlialCAM) and α-crystallin B chain (CRYAB), suggesting a direct role for EBNA1 IgG in neuroimmune mechanisms [13–15]. However, surprisingly little attention has been given to other aspects of EBV-induced antibody responses, particularly given the finding that elevated antibodies to the "EBNA complex"–as defined by immunofluorescence staining of latently-infected cells and comprising EBNA1 plus the co-expressed nuclear antigens EBNAs 2, 3A, 3B and 3C –is linked to a 36-fold increased MS risk [10,11]. This raises the possibility that other latent cycle antigens may also be involved as inducers of pathogenic responses. In that regard, it is worth noting that sequence divergence in EBNAs 2, 3A, 3B and 3C is the main difference between the two types of EBV occurring worldwide [16,17]. Type 1 strains are dominant in most societies, particularly in Europe and US, but it is not fully understood whether there is any skewing of this dominance in MS cohorts.

So far, no consensus has been reached as to whether EBV-specific T cell responses are altered in MS. Again, due to the linkage of EBNA1-specific immunoglobulin G (IgG) with disease, EBNA1 has also been the focus for T cell studies. CD4+ T cell responses to EBNA1 have been found to be increased in frequency and broader in epitope specificity in MS and were also shown to produce cytokines in response to a neuronal antigen pool, fuelling research into cross-reactivity of EBV-specific T cell responses [18]. Further reports of T cells with dual specificity for EBV and self-peptides have emerged in recent years associated with the MS risk allele *HLA-DRB1\*15:01* [19–21], and humanised mouse models which carry *HLA-DRB1\*15:01* have been shown to exhibit higher viral load and poorer virus-specific CD4+ T cell responses [22]. Taken together these findings support the notion of a dysregulated and highly activated EBV-specific T cell compartment which not only has cross-reactivity potential with self-antigens in the CNS, but also an ability to breach the blood-brain barrier and cause damage to the brain

and spinal cord leading to MS. Despite these findings, the extent to which molecular mimicry underpins MS pathogenesis remains a largely unanswered question.

In the present study we characterised antibody responses to EBV in plasma from persons with relapsing-remitting MS (RRMS), healthy controls (HC) and individuals with a recent history of IM (POST-IM), extending analysis from EBNA1 to encompass the other components of the EBNA complex as well as the virus-capsid antigen (VCA)–an immunodominant component of EBV's replicative cycle. In addition to this, *ex vivo* CD4+ and CD8+ T cell responses to EBV were investigated using *in vitro*-transformed autologous LCLs and a panel of overlapping 15-mer peptides comprising the EBNA1 sequence as targets. In addition, the potential cross-reactivity of virus-specific T cell responses–expanded *in vitro* by autologous LCL stimulation–was investigated by screening against autologous B-lymphoblasts infected with a panel of novel Modified Vaccinia Ankara (MVA) viruses engineered to express 9 candidate CNS autoantigens.

## Results

### Study participants

To investigate the cellular and humoral response to EBV in MS, PBMC and plasma samples were collected from 3 groups: pwMS, HC and POST-IM. PwMS had a confirmed diagnosis of relapsing-remitting MS (RRMS) and, at the time of blood donation to the study, were not receiving any steroid or disease modifying treatment nor were they experiencing a clinical relapse (n = 31). Expanded disability status scores (EDSS) and disease duration at the time of sampling were collected from patient records (S1 Table). Most patients were early in their disease course (median 4.5 years, range 0 to 32 years). The median EDSS was 1.0 (range 0 to 6.5).

HC were recruited from lab personnel with no recorded history of IM and were matched with MS donors for age, gender and *HLA-DRB1*15:01* carriage as far as possible (n = 33). Individuals with a recent history of IM (POST-IM) were in the post-convalescent phase and samples were collected 4–6 months following resolution of symptoms (n = 11). Cohort demographics are described in Table 1.

### MS patients have an unchanged viral load in PBMC but increased EBV-specific antibody responses

EBV is carried as a latent (non-virus replicating) infection in circulating B cells and so EBV genome load in PBMC reflects the size of the latently-infected B cell pool. The viral load is high following primary infection or during reactivation of infection but generally stabilises over time to low levels as long-term virus carriage is established. We detected EBV genome copy numbers by quantitative PCR (qPCR) in PBMC from HC, MS and POST-IM donors. There was a wide range of virus loads between individuals in all three groups but median values were clearly elevated in the POST-IM group, consistent with these individuals' recent EBV primary infection (Fig 1A). No significant difference was observed between viral loads from HC and MS groups (Fig 1A) suggesting that there is no overall increase in the number of circulating latently infected B cells in the blood of pwMS compared with HC in the present cohort.

Elevated antibody responses to the EBV latent protein EBNA1 are a well-known phenomenon of MS disease and have been associated with increased odds ratio for development of MS, as well as interacting with environmental risk factors and genetics [3]. We therefore determined plasma IgG levels to EBNA1 in our cohort by enzyme-linked immunosorbent assay (ELISA) and, consistent with previous studies, EBNA1 IgG titres were found to be elevated in MS compared to the HC and POST-IM groups (Fig 1B).

**Table 1. Cohort demographics.**

| Group | No. of donors | Gender (F:M) | Age (years MEAN +/- SD) | *HLA-DRB1*15:01*-positive* | History of IM (%) |
|---|---|---|---|---|---|
| HC | 33 | 2.3:1 | 35.67(+/-9.60) | 15/33 | 0/33 (0%) |
| MS | 31 | 5.2:1 | 36.32 (+/-9.50) | 13/31 | 8/31 (25.8%) |
| POST-IM | 11 | 0.73:1 | 21.42 (+/-3.60) | 6/11 | 11/11 (100%) |

EBV virus capsid antigen (VCA) is an antigenic complex expressed during the lytic cycle and antibody responses to this antigen appear soon after primary EBV infection. VCA IgG are less well investigated in MS literature, but some previous studies report an increase in MS [23,24]. Plasma IgG responses to VCA were determined in our cohort using a semi-quantitative immunofluorescence assay and were observed to be significantly elevated in MS compared to both HC and POST-IM groups (Fig 1C). In contrast, antibody responses to tetanus toxoid IgG were unchanged between MS and HC (Fig 1D) suggesting that this increase is specific to EBV antigens, and not a general humoral dysregulation. A slight increase in tetanus toxoid IgG responses was observed in the POST-IM group compared to HC and MS, however this

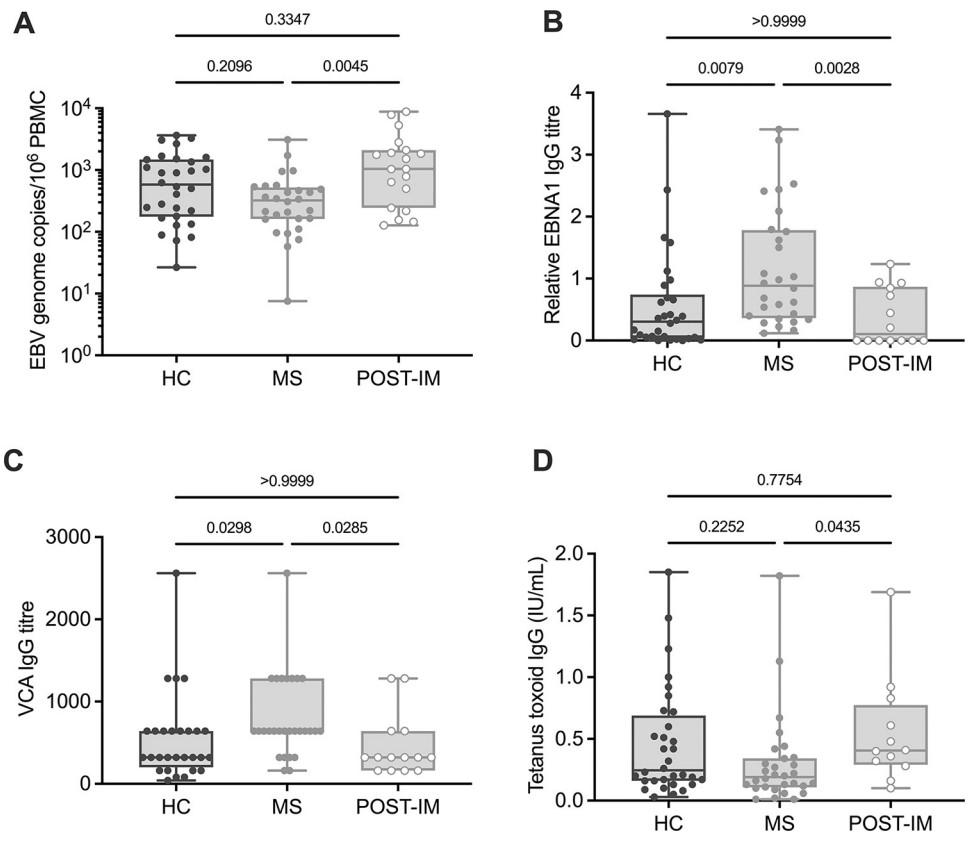

**Fig 1. Viral load and humoral responses to EBV in HC, MS and POST-IM donors. (A)** EBV viral load was determined by qPCR. Viral load was increased in POST-IM PBMC compared to MS (*$p = 0.0045$), but not between HC and MS groups ($p = 0.2096$). **(B)** Plasma anti-EBNA1 IgG was significantly increased in MS compared to HC ($p = 0.0079$) and POST-IM ($p = 0.0028$). **(C)** Anti-VCA IgG titres were significantly increased in MS compared to HC ($p = 0.0298$) and POST-IM ($p = 0.0285$) donors. **(D)** Tetanus toxoid-specific IgG levels were unchanged between MS and HC ($p = 0.2252$) but slightly increased in POST-IM compared to MS ($p = 0.0435$). Horizontal lines represent median of data set. Statistical significance was calculated using the Kruskall-Wallis test with Dunn's post-hoc correction; p-values are reported for each comparison.

could be due to the recent activation of the B cell compartment which occurs during EBV primary infection (Fig 1D). Overall, EBNA1 and VCA IgG responses were both increased in MS compared to HC but tetanus toxoid antibody responses were unchanged. There was no difference in EBV load between inactive MS and HC suggesting the former were not undergoing peripheral EBV reactivation at the time of blood sampling. A significant correlation between EDSS and disease duration was observed (r = 0.374, p = 0.0028; S1 Fig) but there were no significant correlations between EDSS, disease duration, or age and EBV load or antibody responses (EBNA1 IgG, VCA IgG, tetanus IgG, CMV IgG for total cohort or CMV positive individuals only).

## Antibody responses to EBNA2 and EBNA3 latent antigens are more frequent in MS

To gain an overview of antibody responses to all EBV latent proteins, we used the EBNA print Western blot technique which provides semi-quantitative analysis of plasma IgG responses to EBNA1, 2, 3A, 3B and 3C proteins and has been previously described [25,26]. Briefly, this is achieved by performing SDS-PAGE on cell lysates from EBV-negative B lymphoblasts (BJAB) versus EBV-transformed B cell lines and then incubating with diluted human plasma. Antibodies in plasma with specificity for EBV antigens then bind to the membrane at specific molecular weights for each of the different EBNA latent antigens and can be detected with secondary antibodies. Representative EBNA prints from HC, MS and POST-IM individuals are shown in Fig 2A. Confirmation of the bands denoting EBNA1, EBNA2 and the individual EBNA3A, B and C proteins was performed using monoclonal antibodies as previously described [25,26].

Antibodies to EBNA1 are a regular feature of long-term EBV carriage, although very low levels may fall below the limit of detection by Western blotting. In addition, EBNA1 responses can be delayed following IM and may take months to achieve their long-term stable level [27–29]. Consistent with results from the ELISA in Fig 1B, analysis of EBNA prints showed that >96% of individuals with MS had a band corresponding to EBNA1 IgG compared to 80.7% of HC and 86.7% of POST-IM donors (Fig 2C). In contrast antibodies against EBNA2, whether screened by immunofluorescence assays or Western blotting, are typically detectable in 40–50% of healthy carriers [28–30]. In our study we found 40% of both the HC and the POST-IM subjects had detectable EBNA2 reactivity (Fig 2C). Interestingly, that proportion was significantly increased to >60% in our MS subjects, suggesting that anti-EBNA2 responses are more frequent in MS (Fig 2C and Table 2).

Antibody responses to individual antigens of the EBNA3 family were more difficult to resolve using the EBNA print technique due to the similarity in molecular weight of EBNA3A, EBNA3B and EBNA3C proteins. Cell lysates were instead made from the EBV-negative BJAB cell line infected with recombinant MVA viruses engineered to express EBNA3A, B and C antigens of the reference EBV type 1 strain, B95.8 [31]. Western blots were then performed with these cell lysates and plasma to determine the presence of IgG responses to each of the EBNA3 antigens. Representative blots are shown in Fig 2B, with high variation in frequency and intensity of bands corresponding to EBNA3A, B and C IgG observed between individuals and groups. A trend towards increased intensity of bands was observed in plasma from the MS group (Fig 2B) and overall frequency of plasma antibody responses to all EBNA3 proteins were higher in MS than in HC, reaching significance for EBNA3A (HC:MS *p = 0.0220*) and EBNA3B (HC:MS *p = 0.0009*) (Fig 2C). Note that, throughout this work, we also conducted comparative EBNA printing using lysates from a reference type 2 EBV-transformed B cell line and found much lower anti-EBNA2, 3A, 3B and 3C antibody reactivity; this type 1 preference confirmed that the MS patients and comparator groups carried type 1 EBV strains (Table 2) [26].

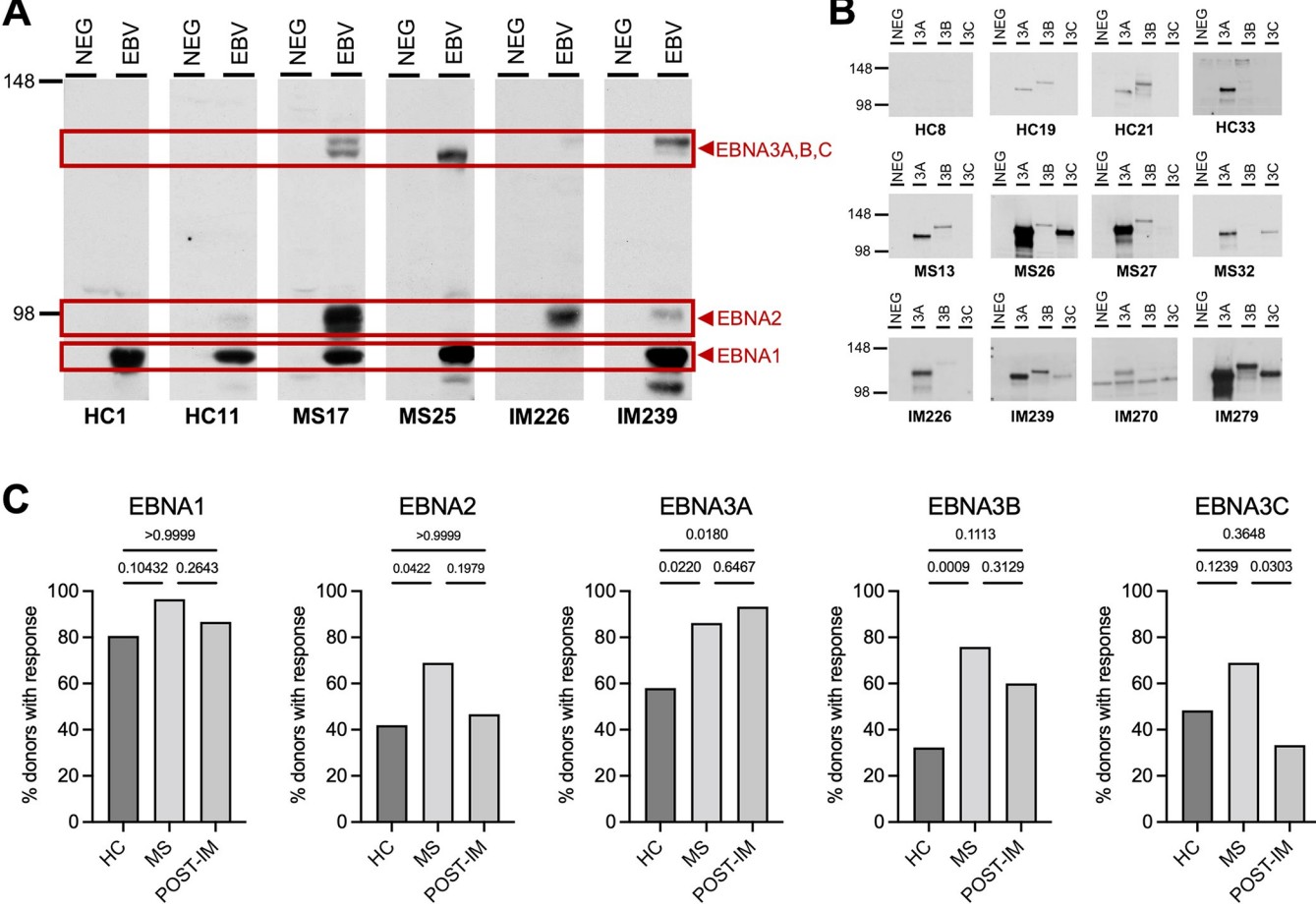

**Fig 2. MS patients show an increased antibody response to EBNA2 and EBNA3 antigens.** Western blots of cell lysates from EBV-negative BJAB cells (NEG) versus EBV-transformed B cell lines (EBV) incubated with human plasma. **(A)** EBNA print IgG analysis of control plasma with arrows indicating bands corresponding to EBNA1, EBNA2 and EBNA3 antigens. Representative EBNA prints from donors are shown. **(B)** Antibody responses were further investigated to EBNA3 proteins using lysate from a cell line infected with individual MVAs expressing EBNA3A, B and C proteins, blots from representative donors are shown. **(C)** Frequency of antibody responses to all EBNA antigens were more frequent in the MS group compared to HC, but were significantly higher for EBNA2, EBNA3A and EBNA3B. Statistical significance was calculated for the data shown in panel C using Fisher's exact test; p-values are reported for each comparison.

Overall, elevated antibody responses to EBV latent antigens were shown to extend beyond the well-established risk factor of high EBNA1 IgG titres in MS, both to VCA and also to EBNA2 and EBNA3 latent antigens. Though only observational, our data indicates a broader

**Table 2. Frequency of EBV latent antigen-specific antibody responses in HC, MS and POST-IM donors.** Donor plasma was probed for antibody reactivity to lysates from C2 cells infected with type 1 (T1) or type 2 (T2) EBV strains by Western blot. Reactivity to EBV latent antigens was determined by bands of the corresponding size as described in the methods section. For analysis of antibody responses to EBNA3 antigens, cell lysates from BJAB cells were infected separately by MVA viruses expressing individual EBNA3A, B and C proteins and antibody reactivity determined by bands corresponding to the correct size.

| Donor | C2 cells infected with type 1 or 2 EBV | | | | | | BJAB cells infected with MVA EBNA3A, B or C | | |
|---|---|---|---|---|---|---|---|---|---|
| | EBNA1 | | EBNA2 | | EBNA3s | | EBNA3 | | |
| | T1 | T2 | T1 | T2 | T1 | T2 | 3A | 3B | 3C |
| HC | 25/31 (80.65%) | 23/31 (74.19%) | 13/31 (41.94%) | 2/31 (6.45%) | 11/31 (35.48%) | 4/31 (12.90%) | 18/31 (58.07%) | 10/31 (32.26%) | 15/31 (48.39%) |
| MS | 28/29 (96.55%) | 28/29 (96.55%) | 20/29 (68.97%) | 6/29 (20.69%) | 21/29 (72.41%) | 14/29 (48.28%) | 25/29 (86.21%) | 22/29 (75.86%) | 20/29 (68.97%) |
| POST-IM | 13/15 (86.67%) | 13/15 (86.67%) | 7/15 (46.67%) | 0/15 (0%) | 7/15 (46.67%) | 5/15 (33.33%) | 14/15 (93.33%) | 9/15 (60%) | 5/15 (33.33%) |

dysregulation of EBV-specific IgG responses in MS than previously acknowledged, which may have mechanistic implications for the EBV humoral response in MS development.

## *Ex vivo* T cell responses to lymphoblastoid cell lines and EBNA1 show subtle differences in frequency and function in MS

Despite the well-characterised elevation of EBNA1-specific IgG in MS, it is unclear to what extent EBNA1-specific or overall cellular immune responses to EBV are altered in MS, with contrasting earlier studies reporting decreased, elevated and also unchanged T cell responses to EBV antigens in MS [32–34]. We therefore sought to investigate the frequency and phenotype of EBV-specific T cell responses in the present cohort by challenging their PBMC *ex vivo* (i) with the autologous LCL transformed with a wild-type EBV strain (WT-LCL) expressing all the latent proteins plus (in a small percentage of cells) the lytic proteins, (ii) with the autologous LCL transformed by a replication-defective BZLF1-knockout version of the same strain [35] therefore expressing only latent proteins (LAT-LCL) and (iii) a pool of overlapping 15-mer peptides spanning the primary sequence of EBNA1. Following overnight *ex vivo* stimulation, PBMC underwent surface and intracellular staining to identify the T cell population and analyse cytokine production of IFNγ, IL-2, IL-17A and GM-CSF (Gating strategy in S2 Fig).

Staphylococcal enterotoxin B (SEB) was used as a positive control stimulus in donors with sufficient PBMC (HC n = 20, MS n = 19). SEB is a superantigen which non-specifically stimulates T cells expressing a particular set of T cell receptors Vβ (TCRVβ) by cross-linking with HLA:peptide complexes and leading to pan T cell stimulation [36]. SEB was used as a non-antigen-dependent stimulus to evaluate cytokine production of T cells between HC and MS donors, and production of IFNγ, IL-2, IL-17A and GM-CSF by responding CD4+ and CD8 + T cells is shown in S3A and S3B Fig respectively. Whilst there were no significant differences in cytokine production in the CD4+ compartment between HC and MS, significantly higher IL-17A production and a trend towards increased GM-CSF production in the CD8+ T cell compartment were observed which likely reflects differences in the TCRVβ repertoires between HC and MS groups (S3A and S3B Fig).

To illustrate the quality of the FACS profiles observed in these experiments, CD4+ and CD8+ T cell responses from one MS donor (MS11) to WT-LCL, LAT-LCL and EBNA1 peptides are shown in Fig 3A and 3B (IFNγ and IL-2). Overall results for IFNγ and IL-2 responses in HC, MS and POST-IM are summarised in Fig 3C and 3D (CD4+ responses) and Fig 3E and 3F (CD8+ responses). High inter-donor variation in responses to each stimulus was observed but some distinctions are apparent. Significantly, higher IL-2 production by CD4+ T cells responding to the LAT-LCL was observed for pwMS compared to POST-IM donors (Fig 3D). Conversely, increased IL-17A production by CD4+ T cells responding to EBNA1 was observed in POST-IM compared to HC and MS (S3C Fig).

The frequencies of CD8+ T cells producing cytokines after WT-LCL and LAT-LCL stimulation were overall much higher than those of CD4+ T cells, with POST-IM donors showing the highest overall responses–consistent with the slow waning of EBV-specific CD8+ T cell numbers in the blood following primary infection (Fig 3E and 3F). However, no significant differences were observed in frequency of IFNγ- or IL-2-producing CD8+ T cells in response to WT-LCL or LAT-LCL between the HC and MS groups (Fig 3E and 3F). CD8+ T cells from MS individuals produced significantly more IL-2 following EBNA1 peptide stimulation compared to the POST-IM group (Fig 3F).

We next analysed production of multiple cytokines by SPICE [37] to understand whether polyfunctionality of responding T cells was altered between groups. T cells responding to all EBV targets showed inter-group heterogeneity of cytokine production in the CD4+

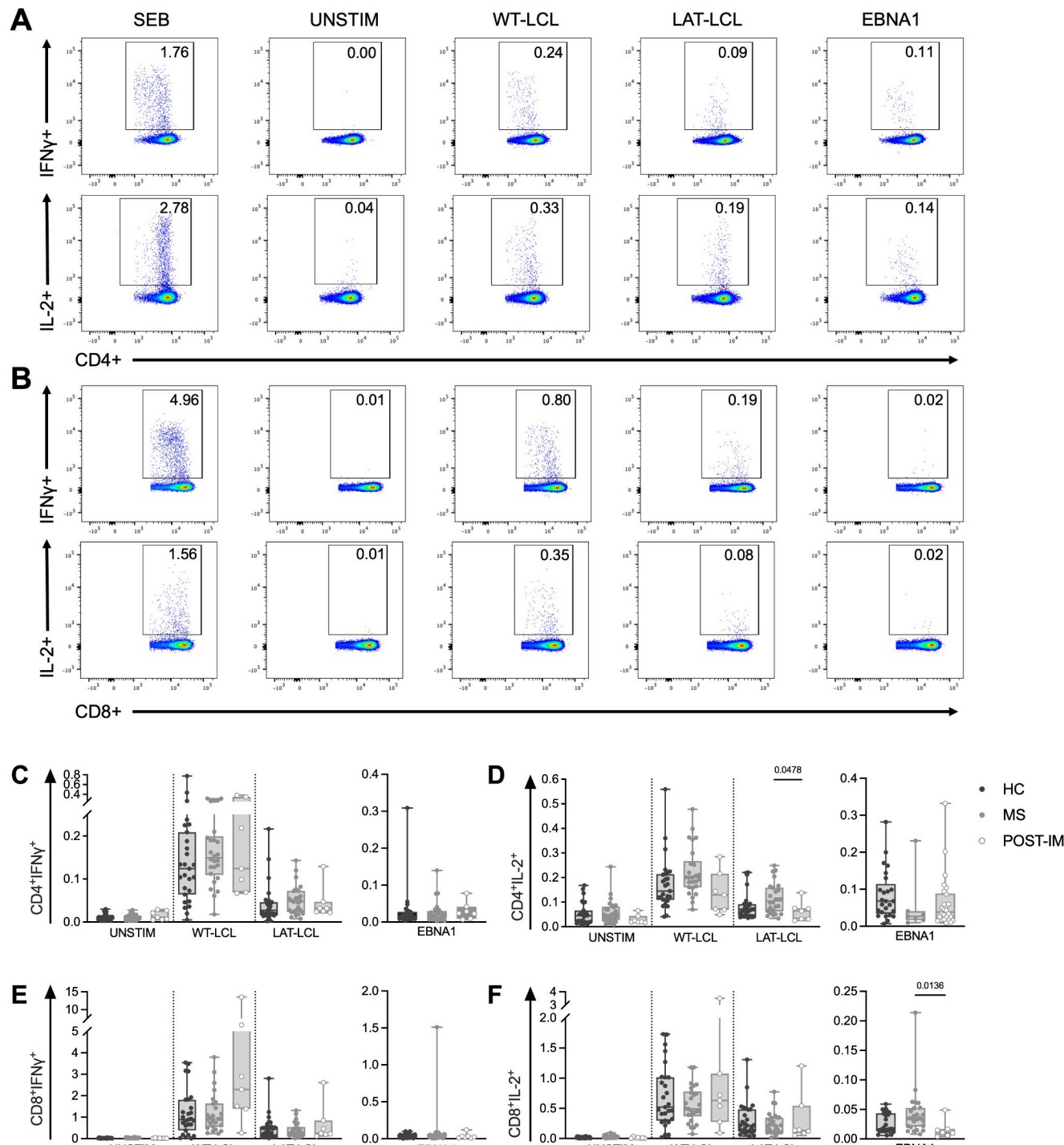

**Fig 3. *Ex vivo* T cell responses to EBV antigens show subtle differences in IFNγ and IL-2 production.** *Ex vivo* PBMC were stimulated with autologous WT-LCL, autologous LAT-LCL or an EBNA1 peptide pool, and CD4+ and CD8+ T cell responses measured by ICS. Example flow cytometry staining of T cells producing IFNγ and IL-2 from one representative MS donor's PBMC (MS11), showing CD4+ (A) and CD8+ (B) subsets. Percentage of CD4+ and CD8+ T cells producing IFNγ and IL-2 after stimulation (C-F). A trend towards elevated IL-2 production CD4+ T cells responding to WT-LCL, LAT-LCL and EBNA1 peptide pool in pwMS was observed compared to HC and POST-IM donors (D) (LAT-LCL IL-2 MS:POST-IM *p = 0.0478*). Increased production of IL-2 was observed in CD8+ T cells responding to EBNA1 peptide pool in pwMS compared to POST-IM donors (F) (EBNA1 stimulation MS: POST-IM *p = 0.0136*). HC n = 27, MS n = 26, POST-IM n = 7. Statistical significance for the data shown in panels C-F was calculated using the Kruskall-Wallis test with Dunn's post-hoc test for multiple comparisons; only significant p-values are shown.

compartment (S4A Fig). In particular, CD4+ T cells from POST-IM donors displayed a different cytokine response profile compared to HC and MS in response to all targets (S4A Fig). CD4+ T cell cytokine profiles were not altered between HC and MS. Interestingly, CD8+ T cell responses were qualitatively less heterogeneous than CD4+, and no significant differences were observed between groups in the cytokine profiles displayed (S4B Fig). Overall, these results indicate alterations in cytokine production specifically in the EBV-specific CD4+ T cell response following primary infection compared to individuals with established long-term virus carriage. Differences in cytokine profiles were also observed for the various stimuli, with LCL stimulation eliciting a response characterised predominantly by IFNγ production, whereas EBNA1 peptide stimulation, produced a greater proportion of IL-2, IL-17A and GM-CSF responding T cells (S4A Fig). Polyfunctionality of cytokine production of by CD4 + and CD8+ T cells following SEB stimulation was unchanged between groups (S4A and S4B Fig).

## EBNA1-specific T cell and antibody responses show modest positive correlation

Whilst EBNA1-specific antibody responses have been studied extensively, less is known about the relationship between cell-mediated and humoral adaptive immune responses to this antigen. We performed linear regression analysis of EBNA1-specific plasma IgG responses and *ex vivo* CD4+ and CD8+ T cell responses to EBNA1 within individuals. The frequency of EBNA1-specific IFNγ+CD4+ T cells was found to have a modest but significant positive correlation with EBNA1-specific IgG titres (r = 0.2620, *p = 0.0450*) when all donor groups were considered together (S5A Fig). Interestingly, EBNA1 IgG titres showed a higher correlation with CD8+IFNγ+ T cells (r = 0.3437, *p = 0.0072*) than CD4+ (S5C Fig). EBNA1-mediated IL-2 +CD8+ T cells in response to EBNA1 stimulation also showed a positive correlation with antibody responses (r = 0.2901, *p = 0.0245*) (S5B Fig). No correlations with any other cytokines were observed.

## LCL-stimulated polyclonal T cell lines show reactivity to neuronal proteins

The mechanism by which EBV contributes towards development of MS is so far not identified, however one striking theory is that T cell responses elicited against EBV are also able to target CNS antigens via molecular mimicry–as has been previously demonstrated for antibody responses [13–15]. To address this possibility, we sought to develop a screening method to detect CNS cross-reactivity within *in vitro*-expanded EBV-reactive T cell preparations. To that end, we used MVA viruses to express nine candidate CNS autoantigens in antigen presenting cells (APC) in a form presentable to both CD4+ and CD8+ T cells.

MVA constructs were made for: 2',3'-cyclic nucleotide 3' phosphodiesterase (CNP), contactin-2, myelin-associated glycoprotein (MAG), myelin oligodendrocyte glycoprotein (MOG), proteolipid protein (PLP), myelin basic protein (MBP), myelin basic protein variant 8 (MBP-V8), myelin-associated oligodendrocyte basic protein (MOBP), alpha-crystallin B (CRYAB), claudin-11 and transaldolase-H (TAL-H). As a safety precaution, MVA viruses were engineered to express these candidate autoantigens under the control of a T7 polymerase promoter. Additional constructs were made expressing EBNA1 (lacking its glycine-alanine repeat sequence, EBNA1ΔGA), an empty vector negative control MVA and also a MVA which expressed bacterial T7 polymerase. Co-infection with MVA T7 polymerase and autoantigen-expressing MVAs enabled autoantigen expression in human cells *in vitro*. All antigen constructs were tagged with an invariant chain sequence to ensure processing of the endogenously-expressed protein via both MHC class I and II pathways in infected cells, in addition to

a FLAG tag to enable detection of protein expression by Western blot (Fig 4A). Recombinant MVA virus titres were calculated by enumerating GFP-positive virus plaques in cells infected at varying multiplicity of infection (MOI) (S6A Fig), and Western blot was used to confirm protein expression and regulation by co-expression of T7 polymerase (S6B Fig). As a source of APCs in the screening experiments, autologous B lymphoblast cultures were established from each donor via CD40:CD40L receptor ligation and IL-4 expansion.

Once these tools were established, we generated EBV-specific polyclonal T cell lines by stimulating PBMC with irradiated autologous WT-LCL on day 0 and day 7 to reactivate EBV-specific memory T cells *in vitro*, after which the cells were expanded in the presence of IL-2. After 28 days, these EBV-enriched polyclonal T cell lines were tested for their potential to cross-recognise CNS autoantigens by challenging with autologous B cell blast cultures infected with individual autoantigen-expressing MVAs (Fig 4B). As positive controls, the LCL-stimulated polyclonal lines were also re-challenged with WT-LCL and LAT-LCL. Cytokine production was measured by ICS and flow cytometry, and IFNγ staining from one representative MS donor for selected targets is shown in Fig 4C.

Throughout these experiments, B cell blasts infected with an MVA-empty vector construct were used as controls and showed that background recognition was generally low in both CD4 + and CD8+ T cell compartments (Fig 4D and 4E). Polyclonal T cell lines from different individuals varied in their ability to recognise the autologous LCL with which they were initially stimulated, likely reflecting the different levels of EBV-specific T cell memory in the blood of these individuals at the time of sampling (S7A and S7B Fig). Strikingly, a high proportion of EBV-specific polyclonal T cell lines from both healthy and MS donors produced cytokines when challenged with MVA-infected B cell blasts expressing candidate autoantigens. Some neuronal proteins emerged as frequently recognised targets for both sets of donors, and included MBP, MOG, PLP, TAL-H and MOBP (Fig 4D and 4E). Unexpectedly, the level of CNS antigen recognition by CD8+ T cells in EBV-stimulated polyclonal lines was generally higher than in CD4+ T cells, a trend that was also consistent between both MS and HC (Fig 4D and 4E). Up to 10% of CD8+ T cells in polyclonal lines were shown to produce cytokine in response to B cell blasts expressing CNS antigens, with each donor displaying a unique pattern of CNS antigen recognition (S7B Fig). Interestingly, recognition of neuronal antigens by EBV-stimulated polyclonal lines by T cells was higher in the HC group compared to MS.

## T cell clones isolated from an MS donor showed dual specificity for EBNA1 and MOG antigens

Due to the high frequency of T cells responding to neuronal proteins in LCL-stimulated polyclonal T cell lines, we hypothesised that this reactivity may be due to structural homology between epitopes from EBV and CNS antigens. We attempted to isolate T cell clones with dual specificity, focusing on reactivity to four MS autoantigens: MOG, PLP, MBP and MOBP. These autoantigens were amongst the most highly recognised in WT-LCL-stimulated polyclonal T cell lines especially in the CD8+ compartment, and previous studies have also showed MBP to contain epitopes which cross-react with EBV [20,21].

Multiple methods of T-cell cloning and cytokine capture were tested including limiting dilution, IFNγ capture and TNFα capture. In these experiments, whole PBMC were stimulated on days 0 and on day 7 either with WT-LCL (as in previous work) or with autoantigen-expressing MVA-infected B cell blasts, and thereafter expanded as polyclonal T cell lines *in vitro*. On day 21, T cells were challenged with the reciprocal stimulus–i.e. with MVA-infected B cell blasts or with WT-LCL respectively–and TNFα capture was performed to isolate the LiveCD3+TNFα+ T cell population reacting to the stimulus (S8 Fig). Note that IFNγ capture

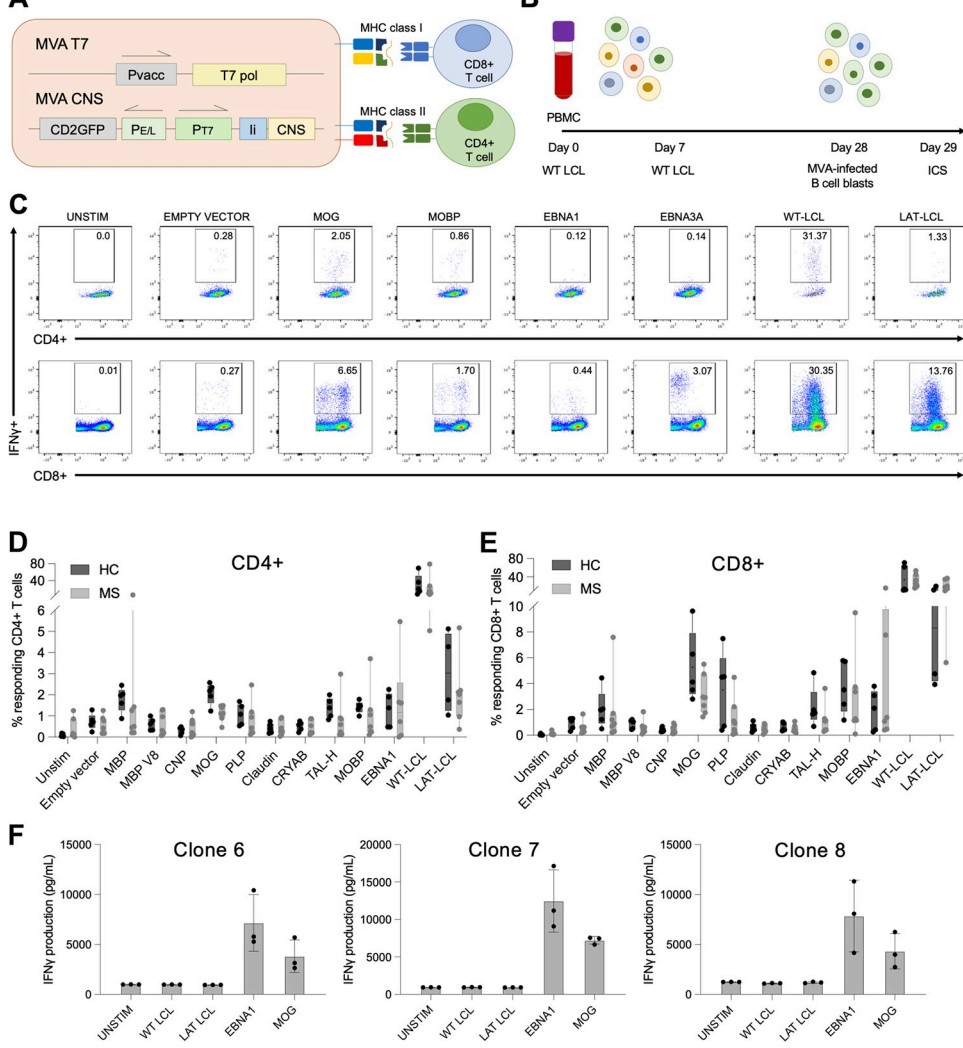

**Fig 4. WT-LCL-stimulated polyclonal T cell lines show high levels of reactivity to CNS antigens. (A)** Schematic of Modified Vaccinia Ankara (MVA-CNS) viruses engineered to express CNS proteins including invariant chain and FLAG tags, under the control of T7 polymerase. **(B)** Whole PBMC were stimulated with autologous WT-LCL on day 0 and day 7, and subsequent polyclonal T cell lines grown out. On day 28, autologous B cell blasts were infected with individual MVA-CNS viruses and then used to stimulate polyclonal T cell lines. **(C)** Representative IFNγ staining of responding CD4+ and CD8+ T cells from one MS donor (MS14) to selected CNS and control antigens. Percentage of CD4+ **(D)** and CD8+ **(E)** T cells responding to CNS antigens producing cytokines (any combination of IFNγ, IL-2, IL-17 and GM-CSF). Boxes show interquartile range and median, HC n = 5 and MS n = 7). **(F)** Individual T cell clones isolated from MS34 that produced IFNγ in response to both EBNA1- and MOG MVA-infected autologous B cell blasts. Dots represent technical replicates from one experiment.

was discounted as a selection method as there is a risk that T cells could capture IFNγ from neighbouring cells which could affect results.

Both the order and type of antigen stimulations were found to have a large impact on experimental outcomes. Importantly, we found that using CNS antigen as the initial stimulus did not yield any clones which subsequently expanded in culture. In contrast, using WT-LCL as the initial stimulus and MVA-infected B cell blasts as the second stimulus generated many clones which could be expanded and tested for dual specificity (Table 3). Screening methods were adapted to take account of the high number of T cell clones, as well as the small numbers

**Table 3. Summary of the frequency and specificity of T cell clones isolated using different sequences of stimulation before TNFα capture.**

| Donor | Initial stimulus | Stimulus prior to TNFα capture | Number of expanded clones | Number of clones with WT-LCL reactivity | Number of clones with CNS reactivity |
|---|---|---|---|---|---|
| HC29 | WT-LCL | MOG | 114 | 54 | 0 |
| MS1 | WT-LCL | MOG | 36 | 11 | 0 |
| MS1 | WT-LCL | MOG/MBP/PLP/MOBP | 36 | 7 | 0 |
| MS1 | WT-LCL | MOG/MBP/PLP/MOBP | 15 | 9 | 0 |
| MS1 | WT-LCL | MOG/MBP/PLP/MOBP | 37 | 11 | 0 |
| MS6 | WT-LCL | MOG | 16 | 3 | 0 |
| MS6 | WT-LCL | MOG | 14 | 2 | 0 |
| MS13 | WT-LCL | MOG/MBP/PLP/MOBP | 4 | 2 | 0 |
| MS13 | WT-LCL | MOG/MBP/PLP/MOBP | 5 | 2 | 0 |
| MS34 | WT-LCL | MOG | 8 | 3 | 4 |
| MS1 | MOG/PLP | MOG/PLP | 2 | 0 | 0 |
| MS13 | MOG/PLP | MOG/PLP | 0 | 0 | 0 |
| MS20 | MOG/MBP/PLP/MOBP | MOG/MBP/PLP/MOBP | 0 | 0 | 0 |
| MS34 | MOG/MBP/PLP/MOBP | MOG/MBP/PLP/MOBP | 0 | 0 | 0 |

of cells per clone that were available for testing. Clones were therefore screened for reactivity to WT-LCL, to LAT-LCL and to autologous B blasts infected with a selection of candidate MVAs; note that, in this situation, we found that IFNγ and TNFα were comparable markers of specific T cell recognition (S9 Fig). Many clones were identified with reactivity to EBV targets, however only 4 T cell clones from one MS donor were found to respond to both types of target. These clones were isolated from a WT-LCL-stimulated polyclonal T cell line from donor MS34 which was re-stimulated with MOG MVA-infected B cell blasts immediately prior to TNFα capture and sorting (Table 3). Interestingly, these T cell clones showed low IFNγ response to the original WT-LCL stimulus, but showed high reactivity to both EBNA1 and MOG antigens (Fig 4F). Three of the T cell clones had low background in the unstimulated control wells but one clone showed very high background even without stimulus and is not shown. All 4 clones produced high levels of IFNγ in response to EBNA1-infected B cell blasts and a lower amount in response to MOG-infected B cell blasts, suggesting that they may have lower avidity for MOG than for EBNA1 (Fig 4F). These extensive experiments investigating dual specificity of EBV-specific T cell clones supports the existence of EBV/CNS autoantigen cross-reactive T cells that have a very limited potential to expand *in vitro*, at least under standard conditions.

## Discussion

The present study aimed to characterise antibody and T cell responses to EBV in MS patients, and to assess the ability of the virus-specific T cell compartment to react to CNS proteins. Through these experiments our aim was to address conflicting data from MS cohorts around the world regarding cellular and humoral EBV immunity. We also sought to include in our analysis not just healthy matched controls but also–given the known increased risk of MS following IM [38]–a set of individuals who had recently recovered from symptomatic primary EBV infection but in whom the virus carrier state has not yet reached its stable life-long position of equilibrium.

Starting with virus load, elevated EBV load in PBMC preparations can signify disturbance of the (usually stable) virus carrier state, which may be due to T cell impairment as is seen in

post-transplant lymphoproliferative disorder or, to a lesser extent, may reflect a dysregulation of the B cell compartment as seen in some autoimmune conditions such as rheumatoid arthritis and systemic lupus erythematosus [39–41]. Unlike these examples, elevations in virus load have only been shown in a minority of MS and clinically isolated syndrome (CIS) studies [33,34,42], with the majority of cohorts showing the overall virus burden to be unchanged when compared to healthy controls in PBMC or plasma [43–46]. Data from our cohorts corroborates most studies' findings of an unchanged EBV load, and therefore argues against uncontrolled virus carriage as a driver of MS pathology. This is further supported by two observations: firstly, individuals with IM do not generally develop neurological symptoms, and secondly, immunocompromised patients, such as human immunodeficiency virus-infected individuals, do not have an increased risk of CNS autoimmunity and are in fact less likely to develop MS [47]. By contrast, rare cases of MS are now being reported in individuals receiving checkpoint inhibitor therapy which breaks T cell tolerance mechanisms to enable the host's immune system to eliminate their cancer, further evidence which supports that MS arises from aberrant T cell autoimmunity rather than uncontrolled infection [48]. IM has been previously shown to increase risk of MS, but this effect is not immediate and instead increases with the severity and the number of years since primary infection [8,49]. This observation suggests that it is not the viral infection itself, but rather the immune dysregulation triggered by acute primary EBV infection, which drives MS development.

Elevated EBNA1 IgG is known to precede MS development and increase disease risk by approximately 3-fold, an effect which also acts synergistically with genetic and environmental factors [3,7,9,50]. EBNA1 is primarily expressed in latently-infected B cells upon cell division where its function is to anchor viral episomes to host chromatin. For reasons that are still not fully understood but may reflect delayed CD4+ T cell help [51], EBNA1-specific IgG does not appear until around 3–6 months post primary infection and thereafter remains stable over time [29]; indeed the presence of VCA IgM and rising VCA IgG titres in the absence of EBNA1-specific IgG are used as the main diagnostic method for IM [52]. Previously suggested to be just biomarkers, there is now evidence that some EBNA1-induced antibodies can cross-react with self-proteins and may have a direct role in damaging the CNS [13–15]. Therefore, elevation of EBNA1 IgG titres in MS patients may be driven by their ability to also target self-antigens and could potentially contribute towards CNS damage. As has also been noted in previous studies, we observed a weak positive correlation between EBNA1-specific antibody and T cell responses in individuals [15]. On the other hand, efficacy of anti-CD20 therapies in MS (such as rituximab and ocrelizumab) is not fully explained by depletion of antibody-producing B cells, as CD20-negative plasma cells are the main source of antibody production, and additionally plasmapheresis is only effective in a relatively small proportion of MS patients and is generally considered ineffective for relapsing-remitting disease [53]. These observations suggest that pathogenic antibodies, and therefore EBNA1 IgG, may not be sufficient to exclusively drive MS pathology, as has also been demonstrated in studies of experimental autoimmune encephalomyelitis (EAE)–the animal model for MS [54].

Antibody responses to lytic proteins and other latent EBV antigens are less well-characterised than EBNA1 IgG in MS, and the present study has identified increased frequency of antibody responses to both EBNA2 and EBNA3 family proteins in pwMS. These data indicate a broader dysregulation of virus-specific antibody responses than previously acknowledged. Such dysregulation was not observed in POST-IM donors, nor was it accompanied in MS individuals by an increased viral load. This suggests that this elevation of antibody responses is not driven by an increased reservoir of infected B cells and does not occur in the months following symptomatic primary infection, but could instead be due to a hitherto unidentified immune mechanism.

Antibody responses to individual EBV latent proteins all show different kinetics following virus acquisition. EBNA2-specific IgG levels peak in the blood alongside viral load at around 25 days following symptomatic primary infection, but less is known about antibody responses to EBNA3A, B and C; indeed these antigens are better known as frequent targets for both CD4 + and CD8+ T cells [27,29,55,56]. Whilst the "EBNA print" method used for analysis of EBNA2 and EBNA3-specific antibody is only semi-quantitative, it nonetheless provides a broad and visually clear idea of the frequency of antibody responses to a wider array of the EBV latent proteins. It is an extremely useful tool given the difficulty in producing these antigens recombinantly and the lack of commercially available serological testing kits for EBNA2 and EBNA3-specific IgG. While it will be important to confirm elevations of EBNA2- and EBNA3-specific IgG in further MS cohorts, our data provides evidence for greater dysregulation of EBV latent protein-specific antibodies than previously thought. Note that dysregulation of latent antibody responses in MS has also been reported in a large prospective study which showed elevations of EBNA1, EBNA2 and early lytic antigen-specific IgG prior to MS disease onset [7]. It is unclear why antibody responses to EBNA2 and EBNA3 proteins occur more frequently in MS patients but–when also considering the observation that virus load is largely unchanged in MS patients–this is unlikely to be due to an increased amount of antigen available to prime IgG responses. Elevated EBNA2- and EBNA3-specific IgG in MS could reflect differences in antigen-processing and presentation in patient B cells, or variations in HLA type, or may simply be biomarkers secondary to as-yet unknown MS disease mechanisms. In any event, our findings reinforce the idea that several EBV proteins, and not just EBNA1 [13–15], may contain epitopes with homology to self-antigens. Further investigations should therefore focus on the ability of these antibodies to bind self-proteins.

Our study shows LCL- and EBNA1-specific T cell responses in MS to be the same if not slightly increased compared to healthy controls, suggesting that there is no deficiency of viral immune control in MS patients as suggested by some studies [34,42]. However, as LCL-stimulations capture the whole EBV-specific T cell repertoire, alterations in responses to individual virus antigens cannot be identified in these experiments. In addition, MS patients in the present study were not currently undergoing clinical relapse and therefore may not have detectable pathogenic cells in peripheral blood; this compartment may also not be truly reflective of immunological events that are occurring in the CNS. It is also interesting to note the generally higher frequency of both CD4+ and CD8+ T cells responding to WT-LCL compared to LAT-LCL across all groups, suggesting a high frequency of T cells targeted against EBV lytic antigens in the blood even in long-term virus carriers.

EBNA1-specific T cell responses have previously been shown to be elevated in PBMC from MS patients, recognising a broader range of epitopes within EBNA1 and were also shown to recognise a pool of myelin antigens [18,57]. The present study showed there to be no overall difference in the frequency of CD4+ and CD8+ T cell responses to an EBNA1 overlapping peptide pool in pwMS compared to HC, but subtle differences in cytokine production were apparent compared to HC and POST-IM donors. Lünemann *et al.* reported EBNA1-specific T cells from MS patients produce IFNγ and IL-2, and our study extends this observation to show that these cells also produce GM-CSF. GM-CSF-producing T cells have been shown to be elevated in MS, and this cytokine is known to characterise a particular subset of pathogenic CXCR4+ T-helper cells which are also reduced after disease modifying therapy for MS [58]. However, previous studies have also demonstrated that pwMS undergoing natalizumab therapy–a drug which prevents lymphocyte tracking to the CNS and gut–have increased frequencies of circulating EBNA1-specific T cells, and therefore it is possible that these cells are generally not located in the blood [15].

Alongside antibody studies, a trend towards altered frequency and cytokine production by LCL- and EBNA1-specific T cell responses linked to MS were here mainly observed in CD4 + T cells, whereas EBNA1-specific CD8+ T cells producing IL-2 were found to be significantly elevated. *In vitro* WT-LCL are activated and transformed B cells which have a latency III phenotype and are reflective of activated latency III B cells found in the tonsils of IM patients early post-infection before their transition to Latency 0 resting B cells; at any one time, a small proportion of cells in LCL cultures will also be spontaneously moving into lytic cycle [59,60]. Therefore, stimulation of *ex vivo* PBMC with autologous WT-LCL in theory will capture the entire repertoire of T cell responses to both latent and lytic virus antigens in individuals. In addition, observations from a previous study showed that LCL-stimulation will also generate classical MHC class II-restricted CD4+ T cells that recognise the LCL yet do not map to any EBV antigen, indeed these CD4+ T cells also recognise certain EBV-negative human B lymphoma cell lines of the correct MHC class II type [61]. This constitutes compelling evidence that self-antigens up-regulated in B cells by EBV transformation can induce a CD4+ T cell response [61]. Supportive evidence also comes tangentially from a mouse model, where ectopic expression of another EBV latent protein LMP1 (a key effector of cell transformation but one that does not induce detectable antibody responses) rendered transformed mouse B cells recognisable by a cytotoxic CD4+ T cell response; once again these CD4+ effectors did not map to any epitope in LMP1 and therefore appeared to be against an LMP1-induced cellular target antigen [62].

Further experiments which challenged EBV-specific polyclonal T cell lines with CNS antigens revealed for the first time the extent to which the virus-induced T cell repertoire can recognise and respond to multiple neuronal antigens. Distinct patterns emerged in polyclonal T cell lines from both pwMS and healthy controls, and polyclonal lines from each donor were shown to recognise at least one CNS autoantigen in the panel. One surprising observation was the high level of recognition of CNS antigens displayed by CD8+ T cells in polyclonal lines, as previous studies have mainly focused on cross-reactivity in the CD4+ T cell compartment due to the association of *HLA-DRB1*15:01* with MS [63–66]. CNS reactivity in the CD8+ T cell repertoire in our experiments may have been captured due to our use of a system that is thought to mimic the type of virally-infected B cells that first initiate and drive the EBV-induced T cell response *in vivo*. It is well known that LCL stimulation *in vitro* elicits strong, cytotoxic responses in which CD8+ T cells are in the majority [55] and, using this type of approach, several previous studies have reported cross-reactivity of EBV-stimulated T cells with neuronal proteins [18,20,21]. Previous studies investigating cross-reactive T cell responses have focused on individual antigens and/or epitopes and often use overlapping peptide pools which may not contain optimal epitopes or the post-translational epitope modifications necessary for maximum T cell recognition. In order to circumvent these potential limitations, our system used autologous LCL and B cell blasts which are completely HLA-matched to each donor and ensured native processing and presentation of CNS proteins in these cells from MVA vectors.

CD4+ T cells in EBV-specific polyclonal lines from patients and healthy controls showed a degree of variation between individuals in their ability to recognise neuronal proteins in the assay which is likely attributable to their different HLA types. CD8+ T cell recognition of neuronal antigens from EBV-driven polyclonal T cell lines from our cohort was even more striking and unexpected, showing a similar pattern to that seen in the CD4+ T cell compartment but at higher frequencies. MBP, MOG, PLP and TAL-H were primarily recognised by CD8+ T cells in LCL-stimulated lines and similar patterns of reactivity were seen in both healthy controls and patients. Whilst CD4+ T cells are classically thought to be the key players in MS pathogenesis due to evidence such as that from EAE models and the strongest genetic association

with HLA-DRB1*15:01, there is also mounting data supporting a role CD8+ T cells. CD8+ T cells are abundant and clonally expanded in MS brain lesions, CNS-resident cells can express low or transient levels of HLA class I and indirect evidence also shows that β-2 microglobulin knockout mice do not develop EAE, suggesting that CD8+ T cells and/or HLA class I antigen presentation are required for disease development [67–69]. EBNA1 –whilst generally a relatively rare target of CD8+ T cell responses due to its glycine-alanine repeat region (GA-repeat) partially limiting its presentation via the MHC class I pathway [70,71]–was also a frequent target in polyclonal lines, with 15.9% of CD8+ T cells in one polyclonal T cell line showing a response to EBNA1. Note that this may partly be explained by the deletion of the GA-repeat region from our MVA construct which would enable more efficient presentation of EBNA1 epitopes in the screening assay. This high degree of reactivity to CNS antigens in LCL-stimulated T cell lines highlights for the first time the vast extent to which EBV infection could prime T cell responses with the ability to target neuronal antigens.

Since these experiments were performed, several new autoantigens in MS have been discovered which, were the study to be designed today, would have been included in the MVA panel. These emerging autoantigens include several neuronal and peripheral proteins such as FABP7, PROK2, SNAP91, RTN3, RASGRP2, GDPLFS, ANO2, GlialCAM and others [13,14,72–74]. Nonetheless, we detected reactivity to several classical MS-associated autoantigens in LCL-stimulated polyclonal lines including MBP, MOG and PLP, suggesting that our experiments captured a proportion of the cross-reactive repertoire despite absence of recently-discovered MS-associated autoantigens in the MVA panel. The increasing number of known autoantigens in MS demonstrates the high level of variability between patients and supports the hypothesis that MS is not caused by one shared autoantigen, but instead likely to arise from multiple avenues of CNS immune attack. This idea is also supported by observations in the present study which showed individuals respond to different combinations of autoantigens at different levels.

We also observed a significant amount of recognition by polyclonal lines derived from HC, with no discernible differences between groups in reactivity to CNS antigens in EBV-driven T cell lines. Neuronal antigen-specific T cell responses have been documented in healthy individuals in previous studies [4], and the present study provides further evidence that this phenomenon is not restricted to pwMS. It is also important to consider that these cells were detectable in the blood and, whilst the extent of CNS cross-reactivity in the EBV-specific T cell repertoire is large in both groups, the differences between HC and MS individuals may instead lie with the ability of these cells to migrate across the blood-brain barrier or other factors such as regulation of their activity by peripheral tolerance mechanisms. T cells are known to drastically change their phenotype *in vitro* and therefore any differences that may exist between dual-specific T cells from HC and pwMS would not be distinguishable after 21 days in cell culture as was used in this study. Future research in this area should therefore analyse the phenotype of these cells directly *ex vivo* and should also aim to investigate cross-reactivity of intrathecal T cell responses, however low cell numbers in this compartment would make these experiments extremely difficult to perform without non-specific expansion.

In summary, the present study has shown that antibody responses to EBNA2 and EBNA3 latent proteins are dysregulated in MS patients in addition to EBNA1 and has also demonstrated for the first time the large potential of the EBV-specific T cell repertoire to target the CNS. This strengthens the hypothesis that EBV's role in development and progression of MS is extremely complex and multifaceted, and may also account for the high level of disease heterogeneity between patients. Greater understanding is needed surrounding EBV's role in CNS autoimmunity, and caution must be taken when designing future therapies for MS which target EBV such as vaccination or adoptive T cell therapies.

## Methods

### Ethics statement

Ethical approval was granted by the West Midlands and Black Country Research Ethics Committee (ethics number 11/WM/0067 for MS patients and healthy control donors; ethics number 07/Q2702/94 for POST-IM donors). All donors provided written, informed consent for the collection of blood samples and their subsequent analysis.

### Study participants

People with MS (pwMS) with a confirmed diagnosis of relapsing-remitting MS (RRMS) were recruited from outpatient neurology clinics at the Queen Elizabeth Hospital, Birmingham and the Guest Hospital, Dudley in the West Midlands, UK. Patients were not receiving any treatment (steroids or disease modifying therapy) or undergoing clinical relapse at the time of blood donation. Expanded disability status scores (EDSS) at the time of sampling were collected from patient records and disease duration was calculated by subtracting the year of first symptom from the year of sampling (S1 Table). HC donors were recruited from laboratory staff who were matched as far as possible with pwMS for age, gender and *HLA-DRB1*15:01* status; all had previously tested positive for EBV-specific antibodies with no verbal medical history of IM. POST-IM donors were recruited from the Queen Elizabeth Hospital Birmingham, UK after a positive heterophile antibody test and blood was donated 4–6 months following resolution of symptoms. All donors were typed for HLA-A, -B, -C, -DR and -DQ alleles by PCR using the service provided by the Anthony Nolan Histocompatibility Laboratories, UK.

### Sample collection and preparation

Peripheral blood mononuclear cells (PBMCs) were isolated from ~50mL donor blood by Ficoll-Paque density gradient centrifugation. PBMCs were subsequently cryopreserved by diluting in RPMI 1640 containing 20% foetal calf serum and 10% dimethyl sulphoxide. Paired plasma samples were collected with PBMC at time of blood draw and stored at -80˚C.

### Target cell lines

Lymphoblastoid cell lines (LCLs) were generated from donors using B95.8 EBV or a replication defective B95.8 recombinant strain lacking the viral lytic protein transactivator gene BZLF1 [35] (LAT-LCL) using RPMI 8% FCS and cyclosporin A (Sigma). B cell blasts were generated *in vitro* from PBMC on irradiated CD40 ligand-expressing mouse L-CD40L cells (L-cells) (ATCC) in Iscove's modified Dulbecco media supplemented with interleukin-4, cyclosporin A and human serum as previously described [75,76]. B blasts were maintained by transferring onto fresh, irradiated L-cells biweekly.

### EBV load

DNA was extracted from $10^6$ PBMC using the Qiagen DNeasy kit following the manufacturer's protocol. DNA was eluted into 100μL buffer and concentration determined using a NanoDrop spectrophotometer. Multiplex qPCR assay was used to determine EBV genome load in samples by simultaneously amplifying the EBV gene BALF5 (viral DNA polymerase) and β2-microglobulin as previously described [77]. 500ng PBMC DNA from each individuals' preparation were tested in quadruplicate and analysed using average of these values which was calculated to give the EBV load per $10^6$ cells. All standards were tested in triplicate.

## EBV and CMV serostatus and tetanus toxoid-specific IgG

Serially diluted plasma was analysed for EBV virus capsid antigen (VCA) IgM and IgG antibodies by indirect immunofluorescence as described previously [78,79]. EBV positive (Akata BL) and negative (Akata-loss BL) control cell lines were used in the IgM assay, and the IgG assay used cell lines P3HR1 and BJAB as EBV positive and negative targets respectively. EBNA1 IgG titres were determined by ELISA (Diamedex) according to the manufacturer's instructions. Tetanus toxoid-specific antibody responses were determined by the Clinical Immunology Service at the University of Birmingham as previously described [80].

## EBNA protein serology

EBNA print analysis was performed using donor plasma as previously described [25,26]. For EBNA3 serology the EBV negative cell line BJAB was infected with recombinant MVA viruses containing EBNA3A, B and C genes separately [61]. BJAB cells were infected at MOI = 10 in RPMI 2.5% foetal calf serum for 24 hours. Cell pellets were washed with PBS, lysed with urea buffer (9M) and sonicated; protein yields were determined using a bicinchoninic acid assay (ThermoFisher Scientific) and bovine serum albumin standards. Lysates were diluted in 2X Laemmli buffer containing β-mercaptoethanol (Bio-Rad) and incubated at 100˚C for 5 minutes. Proteins were separated by SDS-PAGE (acrylamide resolving gels 4–15%, Bio-Rad) using the Mini-PROTEAN electrophoresis system and running buffer (25mM Tris, 0.19M glycine, 0.1% SDS, pH8.3). Resolved proteins were transferred onto PVDF membranes (Trans-Blot Turbo Transfer System) and blocked for 1 hour in PBS-Tween 5% milk solution. Donor plasma was diluted 1:1000 in PBS-Tween 5% milk solution and used to probe membranes for 16 hours at 4˚C. Membranes were washed 5x with PBS-Tween for >1 hour and secondary goat anti-human IgG Fc-specific peroxidase antibody added (1:2000, Sigma) in PBS-Tween 5% milk for 1.5 hours. Membranes were washed 5x over 1 hour with PBS-Tween and bound antibodies were detected using ECL Western blotting detection reagent (Amersham) as per the manufacturer's protocol. Uncropped images of EBNA prints are provided as S1 and S2 Appendices.

## Flow cytometric analysis of EBV-specific T cell populations

Cryopreserved PBMC were thawed and rested overnight in RPMI 8% FCS. PBMC were stimulated with Staphylococcal enterotoxin B (SEB) (0.2μg/mL), autologous WT-LCL (1:1), autologous LAT-LCL (1:1) or EBNA1 peptide mix (JPT) (1μg/mL). Brefeldin A (10 μg/mL) was added after 1 hour and cells were incubated for 16 hours at 37˚C 5% $CO_2$. Cells were washed twice with cold PBS and stained for 30 minutes on ice with surface antibodies: CD3 APC-Cy7 (Cambridge Bioscience), CD4 PE-Cy7 (eBioscience), CD8 PerCP-Cy5.5 (eBioscience), CD14 PECF594 (BD Biosciences), CD19 PECF594 (BD Biosciences) and Live/Dead Red Dead Cell Stain (Life Technologies). Cells were washed with cold PBS and then cold MACS buffer and fixed with 0.4% paraformaldehyde (Sigma) for 30 minutes. Samples were washed twice with cold MACS buffer and permeabilised with 0.5% saponin (Sigma) at room temperature for 10 minutes. Intracellular anti-human antibodies for IFNγ FITC, IL-2 PE, IL-17 Pacific Blue and GM-CSF APC (all from Cambridge Bioscience) were added and incubation was continued for 30 minutes. Samples were washed twice with cold MACS buffer and analysed using an LSRII flow cytometer (BD Biosciences).

## Generation of recombinant Modified Vaccinia Ankara viruses

Recombinant MVA viruses expressing CNS antigens under tight T7 promoter control were generated using pYWK, a modified form of the vaccinia virus shuttle vector pTM1 [81]. In

brief, a rat CD2-GFP reporter gene, expressed under synthetic early/late vaccinia promoter control to allow selection of recombinant viruses, was inserted into pTM1 antisense to the start of the T7 promoter. The first 80 amino acids of the invariant chain, which acts as an endo/lysosomal localisation tag and promotes MHC-I and MHC-II antigen processing [82], were then cloned downstream of the pTM T7 promoter, producing pYWK. This step also introduced unique restriction enzyme sites allowing subsequent insertion of genes of interest downstream and in frame with the invariant chain tag. EBV and CNS genes of interest were then amplified by PCR, using primers designed to incorporate a FLAG tag at the 3' end of each gene, then cloned into the pYWK vector. To express T7 polymerase in trans, the T7 polymerase chain gene from plasmid pTF7-3 [83] was subcloned into vaccinia shuttle pSC11 and used to prepare a recombinant T7 polymerase-expressing MVA virus. All viruses were purified by multiple rounds of plaque purification and seed stocks were prepared for long-term use. All recombinant MVA viruses constitutively expressed green fluorescent protein which was used for calculation of virus titres. Expression of the CNS and EBV genes was achieved by co-infecting cells with the relevant recombinant MVA virus and the T7 polymerase-expressing virus.

### In vitro reactivation protocols

PBMC were re-suspended in RPMI 8% human serum containing IL-7 (1ng/ml) with irradiated (1000rads) autologous WT-LCL (40:1) or MVA-infected autologous B cell blasts (10:1) and incubated at 37˚C 5% $CO_2$ for 7 days. On day 7, cells were stimulated as before and incubated for growth into polyclonal T cell lines for a total of 4 weeks before screening by ICS against MVA-infected B cell blasts; polyclonal lines for T cell cloning were incubated for 2 weeks instead. For T cell cloning, responding cells were isolated by IFNγ-secreting cell enrichment, by TNFα capture or by limiting dilution. IFNγ-secreting cell enrichment was performed according to the manufacturer's protocol (Miltenyi). TNFα capture was performed by adding stimulus to cells in combination with TNFα APC antibody (eBioscience) and TAPI-0 (1μg/ml, Enzo Life Sciences) for 4 hours. 96-well U-bottom plates were prepared for sorting and each well contained 100μL RPMI 8% human serum containing IL-7 (1ng/mL) with feeder cells ($10^6$/mL) and OKT3 (30ng/mL, eBioscience). After 4 hours, stimulated cells were surface stained with antibodies for CD3 FITC (BioLegend), CD19 PECF594 (BD Biosciences) and Live/Dead Aqua (Life Technologies) and LiveCD19-CD3+TNFa+ cells were sorted into prepared plates at 1 cell/well using a FACS Aria II (BD Biosciences). Plates were incubated at 37˚C, 5% $CO_2$ for 5 days, after which 100μL T-cell media (RPMI, 8% foetal calf serum, 1% human serum, 30% filtered MLA-144 (ATCC) supernatant, IL-2 50 IU/mL (Peprotech)) was added to each well. After 2 weeks, microcultures were observed and screened for antigen specificity.

Feeder cells were prepared from PBMC buffy coats (obtained from the Blood Transfusion Service, Birmingham, UK) from three separate donors by Ficoll-Paque density centrifugation separation before mixing together and incubating overnight in RPMI 8% FCS containing phytohaemagglutinin (10μg/mL). The following day cells were washed five times in fresh medium, irradiated (1000rads) and then used in cultures as feeder cells.

### Supporting information

**S1 Table. Disease duration and EDSS at sampling of pwMS.** Disease duration was calculated as the year of sampling minus the year of first reported neurological symptom as indicated in patient journal records. EDSS is the last reported score prior to sampling in patient records. (PDF)

**S1 Fig. Correlation of clinical data with antibody responses in MS.** Antibody responses in individuals were correlated with EDSS, disease duration and age. EDSS is the score reported at the time of sampling, disease duration was calculated as the number of years between first reported neurological symptom and date of sampling, and age is the age of individuals at the time of sampling. Spearman correlation coefficient (r) and significant P values are indicated (* $p < 0.05$). The lines indicate the linear regression slopes and 95% confidence interval of slopes. (PDF)

**S2 Fig. Gating strategy for *ex vivo* PBMC to isolate CD4+ and CD8+ T cells responding to different stimuli.** Ex vivo responding T cells were defined as single, live lymphocytes that were also CD14-CD19-CD3+ before being further divided into CD4+ and CD8+ subpopulations and analysed for cytokine production of IFNγ, IL-2, IL-17 or GMCSF. Data shown is from ex vivo SEB stimulation of donor MS11 PBMC. (PDF)

**S3 Fig. Cytokine production in T cells following *ex vivo* stimulation of PBMC with EBV antigens.** Percentage of total CD4+ (A) and CD8+ (B) T cells producing IFNγ, IL-2, IL-17A and GM-CSF in response to ex vivo stimulation with Staphylococcal enterotoxin B (SEB) by ICS (HC n = 20, MS n = 19). Mann-Whitney test, only significant p-values indicated. Ex vivo PBMC were stimulated with autologous WT-LCL, autologous LAT-LCL or EBNA1 peptide pool, and CD4+ and CD8+ T cell responses measured by ICS. Percentage of CD4+ and CD8 + T cells producing IL-17 and GM-CSF in response to EBV antigens is shown in (C-F). Increased IL-17 production from CD8+ T cells after stimulation was seen in post-IM donors compared to HD and MS groups (CD8+IL17+ EBNA1 HD:IM p = 0.0045, MS:IM p = 0.0134). HD n = 27, MS n = 26, post-IM n = 7. Kruskall-Wallis test with Dunn's multiple comparisons. (* p<0.05, ** p<0.01, *** p<0.001). (PDF)

**S4 Fig. Multiple cytokine production of responding CD4+ and CD8+ T cells.** Tree plot analysis was performed on T cells responding to SEB, autologous WT-LCL, LAT-LCL or EBNA1 peptide pool and data was analysed using SPICE using permutation and Wilcoxon Signed Rank tests. Analysis of CD4+ T cells shown in (A) and of CD8+ T cells shown in (B). (HD n = 27, MS n = 26, post-IM n = 7, * p<0.05, ** p<0.01, *** p<0.001). (PDF)

**S5 Fig. EBNA1-specific T cell and antibody responses are modestly correlated in individuals and all groups.** Plasma EBNA1 IgG responses were correlated with cytokine production by CD4+ and CD8+ T cells responding to ex vivo EBNA1 peptide pool stimulation from each individual. Plasma EBNA1 IgG was analysed by ELISA and values represent the median of at least 3 separate experiments. (A) IFNγ+CD4+ T cells responding to EBNA1 and EBNA1-specific IgG show a significant correlation of r = 2620 (p = 0.0450). (B) IL-2 cytokine production was not found to be significantly correlated. IFNγ+ (C) and IL-2+ (D) CD8+ T cells responding to EBNA1 and EBNA1-specific IgG were also positively and significantly correlated with r = 0.3437 and r = 0.2901 respectively (IFNγ p = 0.0072, IL-2 p = 0.0245). Spearman's rank correlation coefficient was calculated (HC n = 26, MS n = 27, IM n = 7) (* p<0.05, ** p<0.01, *** p<0.001). (PDF)

**S6 Fig. Expression of recombinant CNS proteins in 293 cells using the MVA virus system.** 293 cells were infected with the recombinant MVA viruses expressing CNS antigens, infection was confirmed by expression of green fluorescence protein (GFP) and CNS protein expression

was determined by Western blot using an anti-FLAG tag antibody. (A) Absence of GFP+ plaques in uninfected BHK21 cells and GFP+ plaques in BHK21 cells infected with MVA viruses engineered to express recombinant autoantigens. (B) Western blots showing CNS antigen expression in 293 cells infected with recombinant MVA viruses. Expression of CNS proteins is tightly controlled by T7 polymerase co-expression and antigens were only produced when cells were co-infected with the T7 MVA virus. MOBPv1 was unproductive and the construct was made again; MOBPv2 expressed a band of the expected size. Red arrows indicate CNS protein expression of the expected size.
(PDF)

**S7 Fig. Heatmaps showing individuals' WT-LCL-stimulated polyclonal T cell lines reactivity to CNS antigens.** Heatmap showing individuals' CD4+ (A) and CD8+ (B) T cell reactivity to MVA viruses expressing individual CNS antigens. Data from Fig 3. Gray fill indicates missing data.
(PDF)

**S8 Fig. Gating strategy for single cell sorting of T-cells with potential dual-specificity for EBV and CNS antigens using TNFα capture.** Whole PBMC were stimulated with autologous wild type LCL on day 0 and day 7. At day 21 LCL-stimulated polyclonal T-cell lines were stimulated with autologous B-cell blasts infected with selected MVAs containing CNS antigens in the presence of TAPI-0. Following stimulation T-cells were surface stained and single, LiveCD19-CD3+TNFα+ cells sorted into 96-well plates. Gating strategy is shown for two representative donors (MS13 and MS14); sorted populations from MOG-stimulated B95.8 polyclonal T-cell lines are shown by the black boxes.
(PDF)

**S9 Fig. Production of IFNγ and TNFα in WT-LCL-specific polyclonal T cell lines after stimulation with WT-LCL.** T cells expanded in response to autologous WT-LCL were stimulated on day 20 with WT-LCL before flow cytometry and intracellular cytokine staining (ICS). (A) Example IFNγ and TNFα staining of CD4+ and CD8+ T cells from one donor (MS17) WT-LCL-stimulated polylonal T cell line. (B) A high proportion of CD4+ and CD8+ cells from WT-LCL-stimulated polyclonal T cell lines either produced TNFα alone or co-produced IFNγ and TNFα in response to re-stimulation at day 20 with WT-LCL, indicating that we would be likely to capture a high proportion of antigen-specific T cells by using TNFα capture for T cell cloning (n = 5). (C) The proportion of CD4+ and CD8+ T cells producing different combinations of IFNγ and TNFα after re-stimulation with autologous WT-LCL, data produced from 5 WT LCL-stimulated polyclonal T cell lines.
(PDF)

**S1 Appendix. EBNAprint Western blots.**
(PDF)

**S2 Appendix. EBNA3 MVA Western blots.**
(PDF)

**S1 Data. Participant information and data used to prepare figures.**
(XLSX)

## Author Contributions

**Conceptualization:** Alan Rickinson, Jill M. Brooks, Graham S. Taylor.

**Data curation:** Olivia G. Thomas.

**Formal analysis:** Olivia G. Thomas, Michael R. Douglas, Alan Rickinson, Jill M. Brooks, Graham S. Taylor.

**Funding acquisition:** Alan Rickinson, Jill M. Brooks, Graham S. Taylor.

**Investigation:** Olivia G. Thomas, Tracey A. Haigh, Michael R. Douglas, Alan Rickinson, Jill M. Brooks, Graham S. Taylor.

**Methodology:** Olivia G. Thomas, Tracey A. Haigh, Deborah Croom-Carter, Alison Leese, Yolanda Van Wijck, Jill M. Brooks, Graham S. Taylor.

**Resources:** Michael R. Douglas.

**Supervision:** Alan Rickinson, Jill M. Brooks, Graham S. Taylor.

**Writing – original draft:** Olivia G. Thomas, Alan Rickinson, Jill M. Brooks, Graham S. Taylor.

**Writing – review & editing:** Olivia G. Thomas, Alan Rickinson, Jill M. Brooks, Graham S. Taylor.

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
