## [Decision Letter · Decision Letter 0]

18 Dec 2023

Dear Dr. Taylor

Thank you very much for submitting your manuscript “Heightened Epstein-Barr virus immunity and potential cross-reactivities in multiple sclerosis” for review by PLoS Pathogens. Your manuscript was fully evaluated at the editorial level and by independent peer reviewers. The reviewers appreciated the attention to an important problem, but raised some concerns about the manuscript as it currently stands. These issues must be addressed before we consider a revised version of your study. We cannot, of course, promise publication at that time. We therefore ask you to modify the manuscript according to the review recommendations before we can consider your manuscript for acceptance. Your revisions should address the specific points made by each reviewer.

I am returning your manuscript with three reviews. After reading the reviews and looking at the manuscript, I recommend Major Revision based on the critiques. With additional work, the manuscript will be suitable for a resubmission if you wish to do so. Please pay particular attention to the following reviewer suggestions and give them due consideration:

The titer of EBNA2 and 3s antibodies would be helpful to generate beyond a simple yes/no from the westerns. This would add more to the literature on the levels of anti-EBNA (1, 2, 3s) antibodies that correlate with disease.

Comparing EBV loads during active flares of MS rather than during stable periods.

Adding information about the disease duration in pwMS: Since pwS in this study were all untreated, most of them were likely in the early phases of the disease, which is information of some relevance for disease pathophysiology.

It is unclear whether there is any specific preference for neuronal antigens and whether any antigen expressed in this manner would raise a strong T-cell response. A better negative control would be to express a non-neuronal antigen in the MVA infected B-cell blasts.

Also, as a minor point, please consider the suggestion about including some of the supplemental figures in the main body of the manuscript.

Sincerely,

We cannot make any decision about publication until we have seen the revised manuscript and your response to the reviewers' comments. Your revised manuscript is also likely to be sent to reviewers for further evaluation.

Sincerely,

Italo Tempera

Guest Editor

PLOS Pathogens

Blossom Damania

Section Editor

PLOS Pathogens

Kasturi Haldar

Editor-in-Chief

PLOS Pathogens

orcid.org/0000-0001-5065-158X

Michael Malim

Editor-in-Chief

PLOS Pathogens

orcid.org/0000-0002-7699-2064

Reviewer's Responses to Questions

**Part I - Summary**

Reviewer #1: In this work, the authors evaluated anti-EBV peripheral immune responses in pwMS, healthy

controls and people with recent Infectious mononucleosis. They confirmed an enhanced anti-

EBNA1 antibody production in pwMS and found an increased response against EBNA2 and EBNA3,

too, at least in terms of prevalence. More subtle differences emerged in anti-EBV T-cell responses,

with a modest correlation between CD8+ responses and EBNA1-abs. They also proved dual

specificity of T cell clones in 1 pwMS, with coexistent reactivity against EBNA1 and MOG antigens.

The topic is highly relevant and timely though the questions this paper aims at answering are rather "traditional": the level of originality in trying to provide an explanation for the association between EBV and MS is nor particularlymhigh

Reviewer #2: Epstein-Barr Virus (EBV) infection is strongly associated with Multiple Sclerosis (MS), but the etiological mechanism remains poorly defined. This study aims to further understanding of the the immune responses to EBV in MS patients relative to healthy controls (HCs). To this end, the authors compare a cohort of 33 HC, 31 MS, and 11 post-IM individuals. They find (1) EBV viral load is not elevated in MS patients relative to HCs, (2) antibody response to EBNA1, EBNA2 and EBNA3s are elevated in MS patients relative to His, (3) T-cell responses in MS patients are subtly different than those of HC or post-IM individuals, and (4) EBV-specific T-cell repertoire can target numerous and highly variable CNS autoantigens.

Reviewer #3: This manuscript explores several important aspects of the relationship between EBV and MS. They start with a cohort of age, sex, and HLA-DR15 matched set of healthy and treatment naïve RRMS patients as well as IM patients within 6 months of diagnosis. The study looks at differences in EBV viral load, serology, T cell responses, and cross-reactivity of T cells expanded on EBV-immortalized LCLs with known MS auto-antigens. The authors find that DNA load in the blood of MS patients is statistically similar to that of healthy controls, but lower than IM. This is an important finding (MS/HC similar VL) as there has been some inconsistency in the literature around this question, though this is what most experts believe to be true. Next, they look at serology against EBV proteins and tetanus. They find no difference between the HC and MS groups for tetanus reactivity, but significantly higher EBNA1 and VCA titers in MS as has been previously reported. Here, the authors add to this by studying reactivity to several other EBNA proteins as this hasn’t been well parsed (the historical literature refers to EBNA complex, which really encompasses several nuclear proteins). The authors use westerns detecting EBNA1, 2, and the 3s. In all cases, there is a higher frequency of positivity in MS vs HC, though titers aren’t measured. This concept is important for the field to appreciate, and it may be that some of the other EBNA protein specificities are good biomarkers of disease progression. The authors next turn to T cell responses. They find that CD4+ T cells respond to BZLF1 KO (eg tightly latent) autologous LCLs in MS patients to produce IL-2 significantly more than in HC. However, all other measurements (IFNy in CD4 and IL-2 and IFNy in CD8) in Z+ and ZKO LCLs from HC, MS, and IM were not significantly altered. The CD4+ IL-2 result is interesting, but difficult to interpret. The lack of other associations is likely due to the significant heterogeneity in responses across individuals.

The authors next turn to the question of cross-reactivity of T cell responses between EBV and auto-antigens in MS. They expand polyclonal T cells on LCLs and then ask whether a set of MS auto-antigens are able to stimulate these cells to produce IFNy or IL-2 compared to LCLs or EBNA1 peptide. They find that both HC and MS patient T cells expanded on LCLs are able to respond to several different auto-antigens such as MOG, MBP, and MOBP. This is somewhat surprising and interesting. They also find that these responses could not be elicited by first exposing PBMC to MVA-autoantigen expressing B blasts and then expanding with LCL stimulation. These data suggest one of two possibilities. One is that there are cross-reactive epitopes between EBV proteins and these auto-antigens or that EBV infection somehow supports expression/detection of these antigens in LCLs. While this is an intriguing finding, it remains unclear then what the distinction would be for MS patients as both HC and MS T cells elicit these responses.

**Part II – Major Issues: Key Experiments Required for Acceptance**

Reviewer #1: Points to address

- It would be important to add information about the disease duration in pwMS: since pwS in this study were all untreated, it is likely that most of them were in the early phases of the disease, an information of some relevance for disease pathophysiology..

- Lack of ANO2 (Tengvall, PNAS 2019) and GLIALCAM (Lanz, Nature 2022) and other proteins

in the candidate CNS autoantigens pool (ma viene parzialmetne ammesso nella discussion

L581-588)

- L 369-70 How do the authors explain the enhanced recognition of CNS antigens in HD vs

MS? (anche questo parzialmente discusso L596-605)

Reviewer #2: The goal of the study was to address some conflicting issues in the field regarding EBV immunity in MS, which remains a hot topic in the EBV field. The manuscript is well-written, technically sound and an extensive study of the immune response to EBV and autoantigens in MS patient T-cells. A major strength of the study is the use of autologous LCLs for antigen presentation to measure T cell responses. For the most part, the study confirms and extends previous results describing T-cell responses to EBV antigens. Some of the conclusions may be over-interpreted since the studies are all ex vivo and the cohort of MS patients do not have active disease, and therefore may not demonstrate strong immunological differences from healthy controls. Inclusion of pwMS under active disease may be more informative. Ultimately, many of the conflicting issues in the field remain unresolved, although the authors have done an admirable job in demonstrating the greater complexity of the immunological response to EBV and its potential role in MS pathogenesis.

Specific Comments

Fig. 1. Would be valuable to compare EBV loads during active flares of MS, rather than during stable periods.

Fig 2A. It would be helpful to show the full Western blots, including the low molecular wt fractions, to understand the immunogenicity of LCLs in these patient cohorts.

Fig. 2C should show individual data points in bar graphs.

Fig. 3E should have p-value calculation for the CD8 INFgamma. Also, legend has asterisk for p-values, but there are no asterisks in the Figure.

Fig 4. It is not clear that there is any specific preference for neuronal antigens, and that any antigen expressed in this manner would raise a strong T-cell response. A better negative control would be to express a non-neuronal antigen in the MVA infected B-cell blasts.

Reviewer #3: 1. The titer of EBNA2 and 3s antibodies would be useful to generate beyond a simple yes/no from the westerns. This would add more to the literature around the levels of anti-EBNA (1, 2, 3s) antibodies that correlate with disease.

2. The experiments analyzing cross-reactivity of T cells between EBV LCLs and auto-antigens are interesting, but lack mechanistic depth - perhaps the experiments that indicate that the order of EBV LCL or auto-antigen used to generate the polyclonal responses should be moved from supplement to main tables and figures - these are important/interesting pieces of this puzzle despite not having clear explanations for why these differences exist.

**Part III – Minor Issues: Editorial and Data Presentation Modifications**

Reviewer #1: Points to address

- L 159-160 better “no difference in peripheral EBV reactivation (lytic phase) between

inactive MS and HD”

- L 377 typo “focussing”

- L 439 typo “in infection”

Reviewer #2: Some of the supplemental figures may be more impactful in the main body of the manuscript.

Reviewer #3: As noted above, Supp Table 2 and other Supp Figs should be included in the main part of the paper.

PLOS authors have the option to publish the peer review history of their article (what does this mean?). If published, this will include your full peer review and any attached files.

Reviewer #1: No

Reviewer #2: No

Reviewer #3: No

Figure Files:

Data Requirements:

Reproducibility:

To enhance the reproducibility of your results, we recommend that you deposit your laboratory protocols in protocols.io, where a protocol can be assigned its own identifier (DOI) such that it can be cited independently in the future. Additionally, PLOS ONE offers an option to publish peer-reviewed clinical study protocols. Read more information on sharing protocols at <a href="https://plos.org/protocols?utm_medium=editorial-email&utm_source=a

---

## [Decision Letter · Decision Letter 1]

8 Apr 2024

Dear  Dr. Graham S Taylor,

We are pleased to inform you that your manuscript 'Heightened Epstein-Barr virus immunity and potential cross-reactivities in multiple sclerosis' has been provisionally accepted for publication in PLOS Pathogens.

Best regards,

Italo Tempera

Guest Editor

PLOS Pathogens

Blossom Damania

Section Editor

PLOS Pathogens

Michael Malim

Editor-in-Chief

PLOS Pathogens

orcid.org/0000-0002-7699-2064

Reviewer Comments (if any, and for reference):

Reviewer's Responses to Questions

**Part I - Summary**

Reviewer #1: (No Response)

Reviewer #2: The authors have addressed my previous concerns. The manuscript describes the expanded T-cell immune response to EBV antigens and auto antigens in MS patients relative to HCs and PostIM.

Reviewer #3: The reviewers have responded to my previous concerns.

**Part II – Major Issues: Key Experiments Required for Acceptance**

Reviewer #1: (No Response)

Reviewer #2: (No Response)

Reviewer #3: (No Response)

**Part III – Minor Issues: Editorial and Data Presentation Modifications**

Reviewer #1: (No Response)

Reviewer #2: (No Response)

Reviewer #3: (No Response)

PLOS authors have the option to publish the peer review history of their article (what does this mean?). If published, this will include your full peer review and any attached files.

Reviewer #1: No

Reviewer #2: No

Reviewer #3: **Yes: **Micah Luftig

---

## [Editor Report · Acceptance letter]

14 May 2024

Dear Dr. Taylor,

We are delighted to inform you that your manuscript, "Heightened Epstein-Barr virus immunity and potential cross-reactivities in multiple sclerosis," has been formally accepted for publication in PLOS Pathogens.

Best regards,

Michael Malim

Editor-in-Chief

PLOS Pathogens

orcid.org/0000-0002-7699-2064